# Tuning network topology and vibrational mode localization to achieve ultralow thermal conductivity in amorphous chalcogenides

Kiumars Aryana[1], Derek A. Stewart [2], John T. Gaskins [1], Joyeeta Nag[2], John C. Read[2], David H. Olson[1], Michael K. Grobis[2] & Patrick E. Hopkins [1,3,4✉]

Amorphous chalcogenide alloys are key materials for data storage and energy scavenging applications due to their large non-linearities in optical and electrical properties as well as low vibrational thermal conductivities. Here, we report on a mechanism to suppress the thermal transport in a representative amorphous chalcogenide system, silicon telluride (SiTe), by nearly an order of magnitude via systematically tailoring the cross-linking network among the atoms. As such, we experimentally demonstrate that in fully dense amorphous SiTe the thermal conductivity can be reduced to as low as $0.10 \pm 0.01$ W m$^{-1}$ K$^{-1}$ for high tellurium content with a density nearly twice that of amorphous silicon. Using ab-initio simulations integrated with lattice dynamics, we attribute the ultralow thermal conductivity of SiTe to the suppressed contribution of extended modes of vibration, namely propagons and diffusons. This leads to a large shift in the mobility edge - a factor of five - towards lower frequency and localization of nearly 42% of the modes. This localization is the result of reductions in coordination number and a transition from over-constrained to under-constrained atomic network.

[1] Department of Mechanical and Aerospace Engineering, University of Virginia, Charlottesville, VA, USA. [2] Western Digital Corporation, San Jose, CA, USA. [3] Department of Materials Science and Engineering, University of Virginia, Charlottesville, VA, USA. [4] Department of Physics, University of Virginia, Charlottesville, VA, USA. ✉email: phopkins@virginia.edu

In recent years, there have been numerous efforts to synthesize materials with ultralow thermal conductivities, a crucial parameter in the development of thermoelectric materials, memory devices, and thermally protective coatings[1–5]. It has been generally believed that amorphous solids possess the lowest thermal conductivity possible[6]. The heat transport mechanisms in ultralow thermal conductivity materials are often described using formalisms originally put forth by Einstein, and later refined by others[7–10], which account for some degree of localization of the vibrational modes or strong suppression of vibrational scattering length scales. These concepts, which partially form the basis of analytical minimum thermal conductivity models, are able to successfully predict the thermal conductivity of a wide range of amorphous solids[9].

With advances in nanofabrication, several studies have shown that, by reducing the mean free paths of the vibrational modes, the thermal conductivity of crystalline materials can be significantly lower than this aforementioned minimum limit. For instance, disordered layered and cage-like crystal structures, as well as superatomic clusters with complex unit cells, proved to be an effective approach in increasing the anharmonicity of the system leading to ultralow thermal conductivities[1–4,11–13]. More than a decade ago, Chiritescu et al.[14] found that, by sequentially stacking ultra-thin bilayers of W and Se, the thermal conductivity of $WSe_2$ at room temperature dropped to a record low of ~0.05 $W\,m^{-1}\,K^{-1}$, sixfold lower than the corresponding minimum limit prediction.

In amorphous materials, however, manipulating the atomic structure to reach values below the minimum limit is more complicated, as it is already in its highest disordered state. Thus, a question remains: how can the thermal conductivity of a fully dense amorphous solid be further reduced? One approach is to introduce chemical heterogeneity. For example, in amorphous alloys, in addition to atomic mass mismatch, disruptions in the bond structure or atomic network can significantly impede the propagation of vibrational energy. Prior works have demonstrated that the thermal conductivity of amorphous thin films can be strongly reduced by breaking the number of linkers that connect the atoms through varying the stoichiometry of a given material. For instance, Ghossoub et al.[15] showed that network connectivity in amorphous $CF_x$ can be manipulated by changing the fluorine to carbon ratio, resulting in nearly a factor of two reductions in thermal conductivity. Later, King et al.[16] illustrated that, by introducing hydrogen impurities in amorphous SiC, the connectivity between the atoms transitions from a rigid to a percolated network, resulting in a reduction of thermal conductivity by nearly an order of magnitude. In a similar study, Braun et al.[17] showed that, by altering hydrogen concentration in a-SiC:H and a-SiO:H, the thermal conductivity can be suppressed by a factor of two. In all of these studies, the change in the number of bonds between constituent elements is obtained by introducing an additional impurity, such as fluorine or hydrogen, to the baseline amorphous composition.

The insertion of these impurities not only changes the physical properties of the original material but also introduces chemical complexities to the system. For instance, increasing the hydrogen concentration in a-SiC:H reduces the density by more than a factor of two and also changes the nature of the bonding with the addition of hydrogen-terminated dangling bond sites. These large variations in density and bonding properties are directly related to the associated thermal transport, making it difficult to exclusively pinpoint the effects of bond percolation on thermal conductivity.

Unlike previous studies where the reduction in thermal conductivity is achieved by increasing the scattering rate of heat carriers, in this study, using different compositions of SiTe binary alloys, we demonstrate how vibrational modes can become fully localized to achieve ultralow thermal conductivities. According to our ab initio molecular dynamics (MD) simulations for high tellurium content SiTe, not only is the contribution of propagons to thermal conductivity subdued, but also the mobility edge shifts by a factor of five to lower frequencies, leading to localization of 42% of the modes and ultralow thermal conductivities in these amorphous alloys. In contrast with previous studies, where reductions in thermal conductivity follow reductions in mass density from bond termination and networking, here, depending on the composition of the SiTe system, we achieve ultralow thermal conductivities while increasing the mass density. This is a consequence of the differing network coordinations of silicon and tellurium and resulting vibrational localization that ensues from creating a solid solution of these two differently coordinated elements. Our work not only shows how a large portion of vibrational modes can become fully localized in an amorphous medium, but it also represents a systematic demonstration of how thermal transport changes across different topological network regimes: under-constrained, stress-free, and over-constrained.

In IV–VI and V–VI chalcogenide alloys, as the concentration of one constituent changes relative to another, depending on the number of covalent bonds per element, the mean atomic coordination number changes. This leads to a natural evolution of the atomic network that will directly affect the physical properties of the amorphous alloys. According to glass theory, disordered media are categorized into (i) flexible polymeric glasses consisting of long chains of randomly oriented atoms with low melting temperatures, (ii) stress-free amorphous structures with optimal glass formation properties, and (iii) rigid amorphous solids consisting of a tightly interconnected network of atoms with high melting temperatures. Inspired by Maxwell's mathematical model for truss structures, Phillips[18] proposed that when the number of local bonding constraints, $n_c$, on an atom equals the number of degrees of freedom, the atomic structure is stress-free. For a three-dimensional network, this occurs when $n_c = 3$ and the mean coordination number, $<r_m>$, is 2.40, which is generally referred to as the rigidity threshold; below this rigidity threshold, the material behaves like a polymeric glass, which is classified as under-constrained, whereas above this limit, the material is rigid and classified as over-constrained.

## Results and discussion

To study the effect of the atomic network on the thermal conductivity, we deposited amorphous thin films of silicon telluride, $Si_xTe_{1-x}$, and selenium telluride, $Se_xTe_{1-x}$, with different compositions sandwiched between layers of carbon nitride ($CN_x$) via magnetron sputtering, using both co-sputtering and nano-laminate techniques, as required to achieve the desired composition. We used various spectroscopy techniques such as x-ray fluorescence (XRF), X-ray diffraction (XRD), Raman, and transmission electron microscopy (TEM) to characterize the film compositions, structures, vibrational energies, and thicknesses, respectively (see supplementary note 1). The schematic in Fig. 1a illustrates different possibilities of network topology based on the silicon to tellurium ratio in amorphous SiTe. In chalcogenide glasses, it has been shown that as the structure transitions from an over-constrained to an under-constrained network, additional low-frequency vibrational modes emerge[19–21]. Raman spectroscopy was performed to identify signatures of these additional vibrational modes on samples with different concentrations of tellurium. As presented in Fig. 1b, for thin-film SiTe the spectra look nearly identical for all tellurium concentrations and no considerable shifts in the location of the peak, or the emergence of additional peaks, are detected.

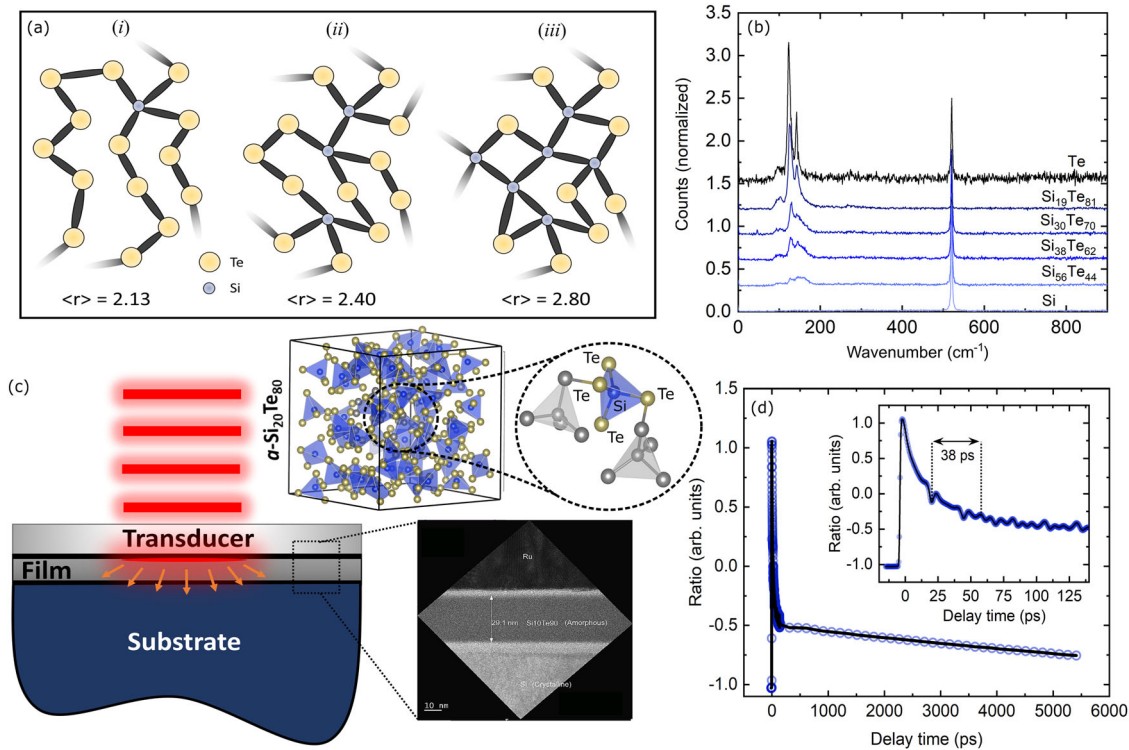

**Fig. 1 The structure and characterization of amorphous SiTe films. a** Schematics of amorphous networks and their corresponding coordination number < $r$ > with a total of 15 atomic sites for different Te to Si ratios, representing various topological regimes: (i) under-constrained network, (ii) stress-free network, and (iii) over-constrained network. **b** Raman spectra for SiTe at different Te concentrations. The sharp silicon peak is the effect of the substrate and is not associated with the thin-film properties. **c** Illustration of the TDTR measurement, a representative atomic structure of $a$-Si$_{20}$Te$_{80}$ with 300 atoms forming randomly oriented tetrahedrons where each Si is bonded to four Te atoms, and a TEM showing the relevant layers in $a$-Si$_{11}$Te$_{89}$. **d** Representative TDTR data for 40 nm thick $a$-Si$_{19}$Te$_{81}$. The inset shows the picosecond acoustics measurements where the distances between the peaks and troughs in the measured TDTR data are related to the time it takes for acoustic waves traveling at the speed of sound to traverse the $a$-Si$_{19}$Te$_{81}$ film.

The cross-plane thermal conductivity and longitudinal sound speed of the SiTe alloys were measured using time-domain thermoreflectance (TDTR), an optical thermometry technique that uses sub-picosecond laser pulses to excite and measure temperature excursions on the surface of materials, enabling the extraction of thermal and elastic properties of thin films[22] (schematic in Fig. 1c). Figure 1d shows an exemplary TDTR thermal decay curve as a function of pump delay time for the 40 nm $a$-Si$_{19}$Te$_{81}$ sample. The inset shows the first few picoseconds of this prototypical TDTR decay curve where the peaks and troughs correspond to the reflection of acoustic waves from the Ru/CN$_x$ and CN$_x$/Si interfaces, respectively. By calculating the time between these "echoes" and with the knowledge of the film thicknesses obtained from TEM, we estimate the longitudinal sound speeds of the SiTe films. The details of sample preparation, measurement technique, and corresponding parameters for data analysis are discussed in supplementary note 2.

In order to understand the effects of atomic network and coordination number on the thermal transport, we consider amorphous SiTe alloys with a variable mean coordination number ($2 < <r_m> < 4$) and amorphous SeTe alloys with a constant mean coordination number ($<r_m> = 2$) at different tellurium concentrations. Figure 2a shows the thermal conductivity of amorphous SiTe and SeTe alloys as a function of Te atomic percentage at room temperature. In SeTe, as shown in red diamonds, the coordination numbers for both Se and Te are identical ($<r_{Te}> = <r_{Se}> = 2$), therefore, the relative atomic concentration does not alter the total bonding network and the thermal conductivity remains relatively constant with increasing

Te content. Thus, in spite of the existence of the large atomic mismatch between Se and Te, the changes in thermal conductivity for these under-constrained amorphous alloys are negligible. To further investigate the effect of atomic mass mismatch on thermal conductivity, we perform MD simulations for amorphous Si with different concentrations of heavy-Si atoms (with atomic mass similar to Te). The inset in Fig. 2a shows the results of these simulations. Similar to what is observed experimentally for the SeTe composition, the thermal conductivity modestly decreases from ~1.0 W m$^{-1}$ K$^{-1}$ in amorphous Si (28 u) to ~0.50 W m$^{-1}$ K$^{-1}$ for $a$-Si with 50% to 90% heavy-Si (127 u) concentrations (see supplementary note 3).

The experimental results for the amorphous SiTe alloys, on the other hand, show that the thermal conductivity of $a$-Si upon mixing with Te drops monotonically by almost an order of magnitude for high Te concentrations. The measured thermal conductivity for our ~26 nm amorphous silicon film with no tellurium is analogous to values reported previously[23] for films of comparable thicknesses ($0.94 \pm 0.2$ W m$^{-1}$ K$^{-1}$). As the Te concentration in SiTe increases, the thermal conductivity drops to as low as ~$0.10 \pm 0.01$ W m$^{-1}$ K$^{-1}$ and stays relatively constant for Te content ranging from 70% to 90%. For the pure Te film, the thermal conductivity increases to $0.23 \pm 0.04$ W m$^{-1}$ K$^{-1}$. TEM micrographs, as well as XRD, reveal that our Te film is not amorphous, but rather polycrystalline. It is not uncommon for Te to crystallize during growth or post-processing as tellurium has a low glass-transition temperature, $T_g$, which makes it difficult to deposit in its amorphous state[24]. The measured thermal conductivity of our Te film is nonetheless lower than those reported in the literature[25–27], which reports in the range of 0.43–3.0 W m$^{-1}$

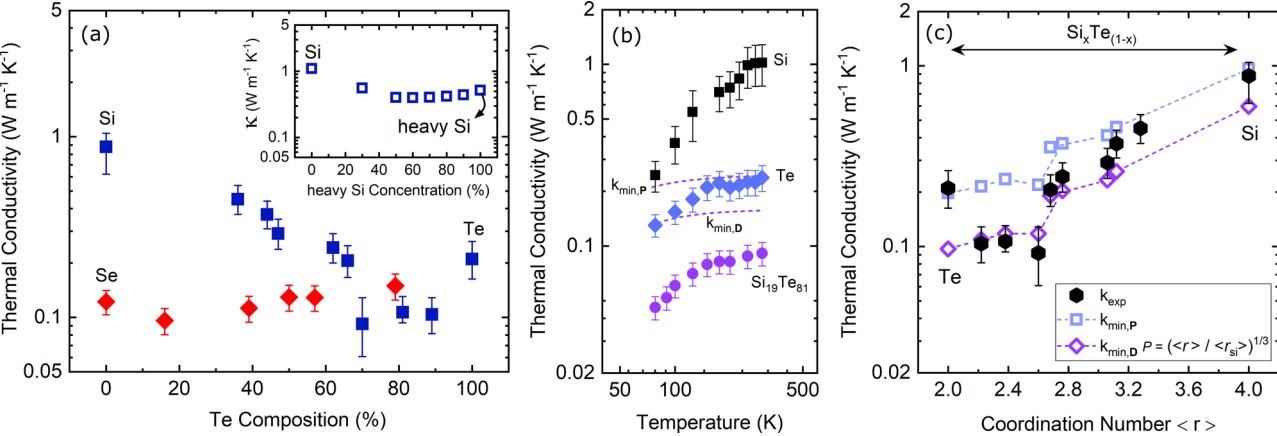

**Fig. 2 Thermal conductivity experimental results and the corresponding models. a** The measured thermal conductivity as a function of tellurium concentration in amorphous SiTe (solid squares) and SeTe (solid diamonds). The inset shows the NEMD simulation results for thermal conductivity of Si (28 u), heavy-Si (127 u), and their alloy. **b** Thermal conductivity of amorphous silicon (solid squares), tellurium (solid diamonds), $Si_{19}Te_{81}$ (solid circles), and minimum limit models for $Si_{19}Te_{81}$ (dashed line) as a function of temperature. **c** Thermal conductivity of SiTe as a function of coordination number measured by TDTR (solid hexagons), their corresponding phonon-mediated (hollow squares), and the diffuson-mediated (hollow diamonds) minimum limit approximations. Uncertainties are calculated based on 10% variations in the thickness of SiTe films.

$K^{-1}$, suggesting that some degree of disorder is present in our Te film. A comprehensive discussion regarding the structure of the films studied here, supported by TEM and XRD measurements, is given in supplementary note 1.

Cryogenic thermal conductivity measurements shown in Fig. 2b reveal that the thermal conductivities of pure Te and a-SiTe samples follow an amorphous-like trend, which is an indication of high levels of disorder in the films. This further explains why the thermal conductivity of the polycrystalline Te film is lower than previous literature values. In all of the cases studied here, the temperature-dependent thermal conductivity trends follow those of the heat capacity and plateau above the Debye temperature.

Although increasing the heavy element Te content in the alloy and the large atomic mass mismatch between Si and Te could partly explain the reductions observed in thermal conductivity, we show they are not the primary reason behind such a dramatic reduction in thermal conductivity. In order to understand the reduction in thermal conductivity of the Si upon the addition of Te, we use existing theoretical approaches to calculate the minimum limit of thermal conductivity derived in the context of phonons ($\kappa_{min,P}$) and diffusons ($\kappa_{min,D}$). According to Cahill and Pohl[28], a lower limit to the thermal conductivity of materials is estimated based on the collective atomic vibrations, i.e., phonons, derived from the kinetic theory of gases:

$$\kappa_{min,P} = 1.21\, k_B\, n^{2/3}\, v_g, \tag{1}$$

where $k_B$ is the Boltzmann constant, $v_g$ is the average sound velocity of the material, and $n$ is the number density. The average sound velocity can be written in terms of the longitudinal ($v_{LA}$) and transverse ($v_{TA}$) sound velocities as: $v_g^2 = \frac{1}{3}\left(v_{TA1}^2 + v_{TA2}^2 + v_{LA}^2\right)$. Although this minimum limit to thermal conductivity has served as a successful approach to predict the thermal conductivity of disordered crystals and amorphous materials, several recent works have experimentally measured values well below this limit. This has motivated others to model the thermal conductivity in amorphous solids as a form of energy hopping between localized vibrational eigenstates. According to Allen and Feldman[29] (AF), a large portion of heat in disordered solids is transferred by harmonic coupling of quantized vibrations that are neither propagating nor localized. These delocalized non-propagating vibrational modes, *diffusons*, carry heat by interactions with other vibrational modes with length

scales on the order of the vibrational wavelength. According to AF theory, the thermal conductivity due to diffuson contribution can be approximated by:

$$\kappa_{AF} = \frac{1}{V}\sum_{i=1}^{N} C(\omega_i)\, D(\omega_i), \tag{2}$$

where $V$ is the system volume, $\omega_i$ is frequency of $i$th mode, and $C(\omega_i)$ and $D(\omega_i)$ are the frequency-dependent specific heat and mode diffusivity, respectively. Based on the AF formalism, Agne et al.[9] suggested a modified minimum limit model for heat transport in disordered solids that relies on the concept of diffusons rather than phonons. They argued that, in a disordered solid, the lower bound to thermal conductivity occurs when thermal transport is entirely driven by diffusons. This approach, albeit with heat transfer carrier length scales being fundamentally different from those modeled in Eq. 1, leads to a similar functional form for the thermal conductivity of disordered materials:

$$\kappa_{min,D} \approx 0.76\, P\, k_B\, n^{2/3}\, v_g, \tag{3}$$

where $P$ is the probability of successful energy transfer between the atoms. The dashed lines in Fig. 2b show the theoretical minimum limits for $a$-$Si_{19}Te_{81}$ based on phonon- and diffuson-mediated thermal conductivity. In the high-temperature limit, and maximum diffusivity where $P = 1$, the calculated diffuson-mediated thermal conductivity is ~37% lower than the phonon minimum limit model. Figure 2b demonstrates that the measured thermal conductivity of $a$-$Si_{19}Te_{81}$ is well below the minimum limit calculations for both existing models. This implies that the thermal transport mechanism in $Si_{19}Te_{81}$ is dominated by other atomistic properties that impede the transfer of energy beyond those accounted for in the minimum limit models.

To resolve this, we revisit an assumption that was made in the diffuson-mediated thermal conductivity, which assumes the successful transfer of energy between diffusons is 100%. As discussed earlier, since the coordination number in SiTe decreases by increasing the Te concentration, the alloy transitions from an over-constrained to an under-constrained network. This reduction in the number of bonds per atom eliminates the number of pathways through which diffusons can interact, and leads to a reduction in the probability of their successful energy transfer. Figure 2c shows the thermal conductivity of our SiTe sample as a function of coordination number. We calculate the coordination number of these alloys based on the measured

relative atomic percentages assuming Si and Te as four- and two-coordinate elements, respectively. The results of our calculation for $\kappa_{min,P}$ and $\kappa_{min,D}$ at different coordination numbers are depicted in Fig. 2c with hollow squares and hollow diamonds, respectively. In this case, the calculation of $\kappa_{min,D}$ assumes that $P$ changes with respect to the mean coordination number of each composition. Following the model proposed by Xi et al.[30] showing a power-law dependence between mean coordination number and thermal conductivity ($k \propto \langle r_m \rangle^{1/3}$), we normalize each coordination number with respect to pure Si, i.e., $P = (\langle r_m \rangle / \langle r_{max} \rangle)^{1/3} = (\langle r_{Si_x Te_{1-x}} \rangle / \langle r_{Si} \rangle)^{1/3}$. Using this assumption, $P$ changes from 1 for Si ($\langle r_{Si} \rangle = 4$) to ~0.8 for Te ($\langle r_{Te} \rangle = 2$). Applying this condition, we calculate the diffuson-mediated thermal conductivity for $Si_{19}Te_{81}$ as 0.12 W m$^{-1}$ K$^{-1}$, in better agreement with the measured values for our SiTe alloys (see supplementary note 4). These results imply that the nature of the vibrational modes contributing to thermal conductivity in these amorphous films becomes more spatially localized as the coordination number is reduced and the SiTe atomic network transitions away from the over-constrained regime.

Although the theoretical models discussed above provide reasonable qualitative insight into the thermal transport mechanisms in the amorphous SiTe alloys, we turn to ab initio MD and lattice dynamics simulations to gain a deeper understanding of the mechanism that drives the ultralow thermal conductivity of $Si_{19}Te_{81}$. In a-Si, Seyf and Henry[31] showed that a small defect concentration (~10%) leads to a dramatic decrease in the population of propagons. Given the large concentration of Te in a-$Si_{19}Te_{81}$, we assume that low-frequency propagons will have a negligible contribution to the total thermal conductivity and focus our attention on medium-frequency heat-carrying diffusons and high-frequency locons (fully localized modes). Locons are vibrational excitations with vibrational amplitudes that decay exponentially from the center of excitation, and normally have high frequencies. The frequency above which vibrational modes are fully localized is known as the mobility edge. The inverse participation ratio (IPR) determines the degree of localization of modes and is given by:

$$IPR = \frac{\Sigma_i^N \left( \Sigma_{\alpha=1}^3 u_{i\alpha}^2 \right)^2}{\left( \Sigma_{i=1}^N \left( \Sigma_{\alpha=1}^3 u_{i\alpha}^2 \right) \right)^2}, \qquad (4)$$

where $N$ is the number of atoms and $u_{i\alpha}$ is the eigenvector component for atom $i$ in the direction $\alpha$. For a mode that is fully localized on a single atomic site, IPR = 1, and for a fully delocalized mode that spans all atoms, IPR = 1/N. Although it is not possible to define a precise IPR value for the transition of modes from diffusons to locons, for this work, we define locons as modes where the eigenvector is spread across 20% or less of the atoms in the supercell (60 atoms, IPR ≥ 0.01667). This convention has been used in a number of previous studies[32–34] and serves as a reasonable cutoff for comparison with other works.

Figure 3a–c show the vibrational density of states (DOS), mode diffusivity, and IPR, respectively, for amorphous Si. The thermal conductivity in a-Si is dominated by low-frequency propagons and diffusons. However, since we assume negligible contribution of propagons to thermal conductivity of SiTe films with high Te contents, we only focus on the diffusons contribution which is ~0.88 W m$^{-1}$ K$^{-1}$ at room temperature for a-Si. Based on the calculated mobility edge, we find that locons make up <4% of modes in a-Si, in agreement with the previous studies[35]. This indicates that almost all modes in a-Si are active in transferring heat. The small percentage of locons could be a result of the over-constrained bonding network in a-Si.

To assess our assumption that coordination number, i.e., number of constraints on an atom, can lead to localization, we turn to another familiar amorphous structure, a-$SiO_2$. The mean

coordination number for this structure is 2.67, which is lower than a-Si and, based on our assumption, should have a larger percentage of locons than a-Si. This composition also has a similar coordination number and atomic structure to a-$SiTe_2$. For instance, the a-$SiO_2$ structure consists of randomly oriented tetrahedrons where each Si is bonded to four oxygen atoms and each tetrahedron shares an oxygen atom with another tetrahedron (Si-O-Si). Given these similarities, a major difference between a-$SiO_2$ and a-$SiTe_2$ is the large atomic mass difference between oxygen and tellurium. In order to facilitate comparisons between a-$SiO_2$ and a-$SiTe_2$ we replace the oxygen mass ($m_O = 16$ amu) with heavy-oxygen ($m_{Te} = 127$ amu). We use a a-$Si^{127}O_2$ taxonomy for our modified $SiO_2$ to avoid confusion with the real system.

Another notable difference between a-$SiO_2$ and a-$SiTe_2$ systems is the bond enthalpy, which is significantly higher for Si-O (~450 kJ/mol) compared with SiTe (~220 kJ/mol) and Si-Si (~225 kJ/mol)[36]. This indicates that strong Si-O bonds would favor high-frequency modes. As can be seen in Fig. 3d, the a-$Si^{127}O_2$ structure has modes with frequencies higher than a-Si. We attribute the emergence of these higher frequency modes to the artifact of an improper interatomic potential. As a result of this, the Si-$^{127}$O bonds in our modified a-$SiO_2$ system are shorter than SiTe, leading to the appearance of high-frequency modes observed in DOS calculations. Nevertheless, these high-frequency modes are localized and do not contribute to thermal conductivity. Figure 3f shows that the mobility edge is shifted to ~400 cm$^{-1}$, lower frequency compared with a-Si, and the percentage of locons has increased to ~10%. The AF thermal conductivity for a-$Si^{127}O_2$ system is estimated to be ~0.55 W m$^{-1}$ K$^{-1}$.

In order to further investigate the effect of coordination number on the localization of modes, we consider $Si_{20}Te_{80}$ composition with a coordination number of 2.4 at the rigidity threshold. For this structure, we assume that a-$Si_{20}Te_{80}$ consists of randomly oriented SiTe tetrahedrons where the tetrahedrons do not share a Te atom (Si-Te-Te-Si). Using force constants derived from ab initio calculations, we report the DOS, diffusivity, and IPR for a-$Si_{20}Te_{80}$ alloy in Fig. 3g–i. According to our IPR calculations, in this composition not only is the effective frequency range reduced but also ~42% of the modes are localized. We find that locons begin to appear at frequencies >89 cm$^{-1}$ and above 102 cm$^{-1}$ all vibrational modes act as locons and do not contribute to thermal transport. This includes the large vibrational peak centered at 125 cm$^{-1}$ owing to tellurium motion and the higher optical band from 250 cm$^{-1}$ to 400 cm$^{-1}$ primarily attributed to silicon atoms. The IPR calculations show the mobility edge is shifted from 550 to 102 cm$^{-1}$, a shift toward lower frequencies by more than a factor of five from a-Si to a-$Si_{20}Te_{80}$. We calculate the AF thermal conductivity without any fitting parameters in a-$Si_{20}Te_{80}$ as 0.10 ± 0.005 W m$^{-1}$ K$^{-1}$, in excellent agreement with the experimentally measured value. Uncertainty is calculated by changing the broadening factor by 50%.

Figure 4a–c show the thermal conductivity accumulation as a function of vibrational frequency at 300 K. The calculated thermal conductivity for a-Si is comparable to that calculated by Larkin and McGaughey[37]. We observe that, for all cases, beyond the mobility edge limit where the modes are fully localized, the thermal conductivity stays constant. This is expected since locons contribution to total thermal conductivity is negligible. This also indicates that for a-$Si_{20}Te_{80}$, the large number of localized modes (42%), leads to ultralow thermal conductivity. In order to visualize these processes, the spatial projection of eigenvectors on a $xy$-plane for delocalized and localized modes are plotted in Figs. 4d, e at frequencies below (41.4 and 41.6 cm$^{-1}$) and above (240.4 and 242.5 cm$^{-1}$) the mobility edge. According to AF

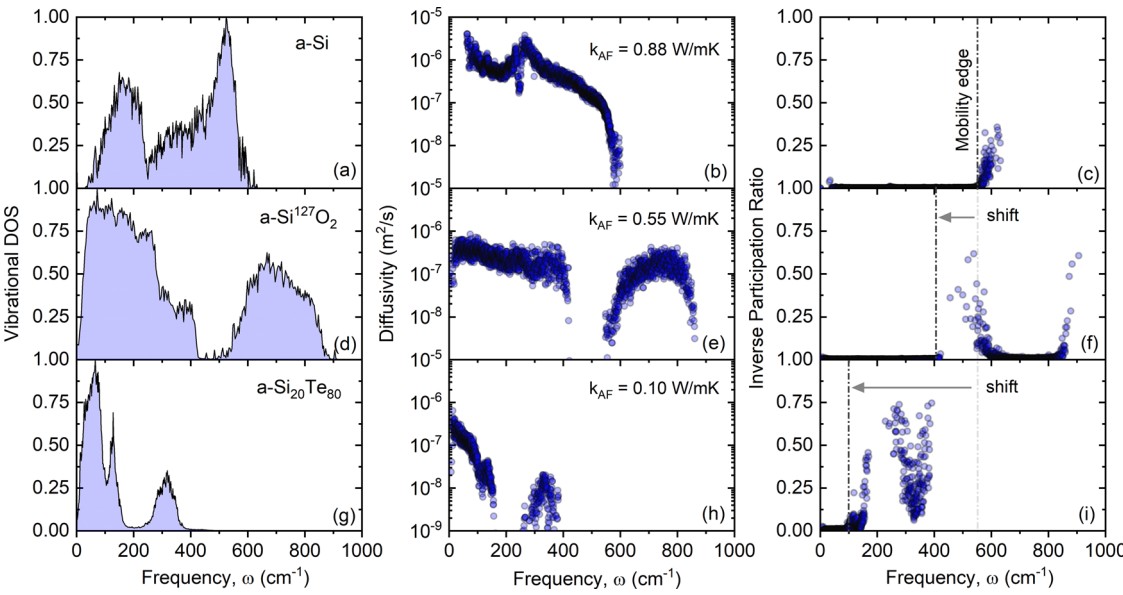

**Fig. 3 Ab initio simulations results for vibrational modes in amorphous Si and its alloys. a, d, g** Vibrational density of states DOS, **b, e, h** modes diffusivity, and **c, f, i** inverse participation ratio (IPR) as a function of frequency for amorphous silicon, $a$-Si, amorphous silicon heavy-oxide, $a$-Si$^{127}$O$_2$, and amorphous silicon telluride, $a$-Si$_{20}$Te$_{80}$, respectively. The corresponding thermal conductivity is given for each system. The shift in the mobility edge with respect to the $a$-Si is depicted by an arrow.

theory, the degree of energy hopping between the modes depends on their spatial (eigenvectors) and energetic (frequencies) overlap. At delocalized frequencies, Fig. 4d, not only do the eigenvectors overlap but also the energy of the modes (frequency) is very close, leading to large diffusivities. On the other hand, in Fig. 4e, although the energy of the modes is relatively close, a lack of spatial overlap prevents the modes from coupling with one another leading to small diffusivities and localizations[38]. In addition, the amplitude of eigenvectors at localized frequencies is strongly suppressed compared to delocalized modes. The large amplitude associated with some of the eigenvectors in Fig. 4e is indicative of strong localization showing the energy associated with these modes is confined in a small geometric region. Figure 4f, g indicate the population of eigenvectors based on their amplitude. According to these figures, for a frequency below the mobility edge, the amplitude of the eigenvectors is uniformly spread out from 0 to ~15. Whereas for frequencies above the mobility edge, due to the effect of localization, the amplitude of the majority of modes is below ~2.

The thermal properties of chalcogenide materials across different topological constraint regimes have been previously investigated for bulk silicon telluride and arsenic selenide[39,40]. In stark contrast to the results presented in this study, the authors observe a peak for thermal diffusivity and thermal conductivity at the rigidity threshold. Philip and Madhusoodanan[39] reported a thermal diffusivity of ~0.06 cm²/s for bulk $a$-Si$_{20}$Te$_{80}$, which is more than a factor two higher than that of amorphous silicon[41]. This could indicate that either their sample is not entirely amorphous or there is a large concentration of impurities. This discrepancy between the bulk and thin films, however, is not surprising as the defects such as impurities are common in bulk systems.

A close material cousin to SiTe is the well-known thermoelectric/phase-change material, GeTe, which has been extensively studied before both in terms of electrical and thermal properties. Although several studies reported the thermal conductivity of GeTe in the amorphous phase at different tellurium concentrations[42–45], depending on the deposition process and thermometery technique,

the values range from 0.1–0.23 W m⁻¹ K⁻¹. The absence of a unique investigation on the thermal properties of GeTe with respect to the coordination number makes it difficult to compare any trend in this composition against that of SiTe. However, owing to the structural similarity between SiTe and GeTe, we use our model for $a$-Si$_{20}$Te$_{80}$ and replace the Si atomic mass with that of Ge ($a$-$^{78}$Si$_{20}$Te$_{80}$) and calculate its diffusivity and thermal conductivity. Since GeTe has a higher average atomic mass, one would expect to observe a lower thermal conductivity compared with SiTe. However, according to our calculations, the thermal conductivity for heavier $a$-$^{78}$Si$_{20}$Te$_{80}$ does not change and remains similar to $a$-Si$_{20}$Te$_{80}$. This demonstrates that the low-frequency heat-carrying modes are not influenced by mass scattering in these unique topological chalcogenide phases.

In summary, we experimentally demonstrate that, through manipulation of the coordination number in amorphous silicon telluride (SiTe) alloys, the thermal conductivity can reach an ultralow value of ~0.1 W m⁻¹ K⁻¹. We observe that the thermal conductivity decreases with coordination number in SiTe and reaches its minimum near the rigidity threshold where the coordination number is 2.2–2.6 with tellurium concentration of 90–70%. We attribute the ultralow thermal conductivity of $a$-Si$_{20}$Te$_{80}$ to the strong localization of heat-carrying modes evident by a large shift in mobility edge—a factor of five—towards lower frequencies.

## Methods

**Experimental approach.** Thermal conductivity measurements were performed via TDTR, a pump and probe optical thermometry technique that relates changes in the thermoreflectance of a sample to thermal properties. In our two-tint TDTR configuration, the output of an 80 MHz Ti-Sapphire femtosecond laser pulse with center wavelength of 808 nm is split into a pump and a probe path. The pump path is directed to an electro-optical modulator (EOM) with modulation frequency of 8.4 MHz, and the probe path is directed to a mechanical delay stage to capture the changes in temperature of the sample as a function of time. The femtosecond laser pulses create an oscillatory temperature rise (a few Kelvin) on the surface, which induces changes in the thermoreflectance of an 80 nm ruthenium layer deposited on the surface of the sample that can be related to the thermal properties of the underlying layers. All samples were coated with an 80 nm ruthenium transducer. The pump and probe beams, after passing through a ×10 objective, focus to spot sizes of

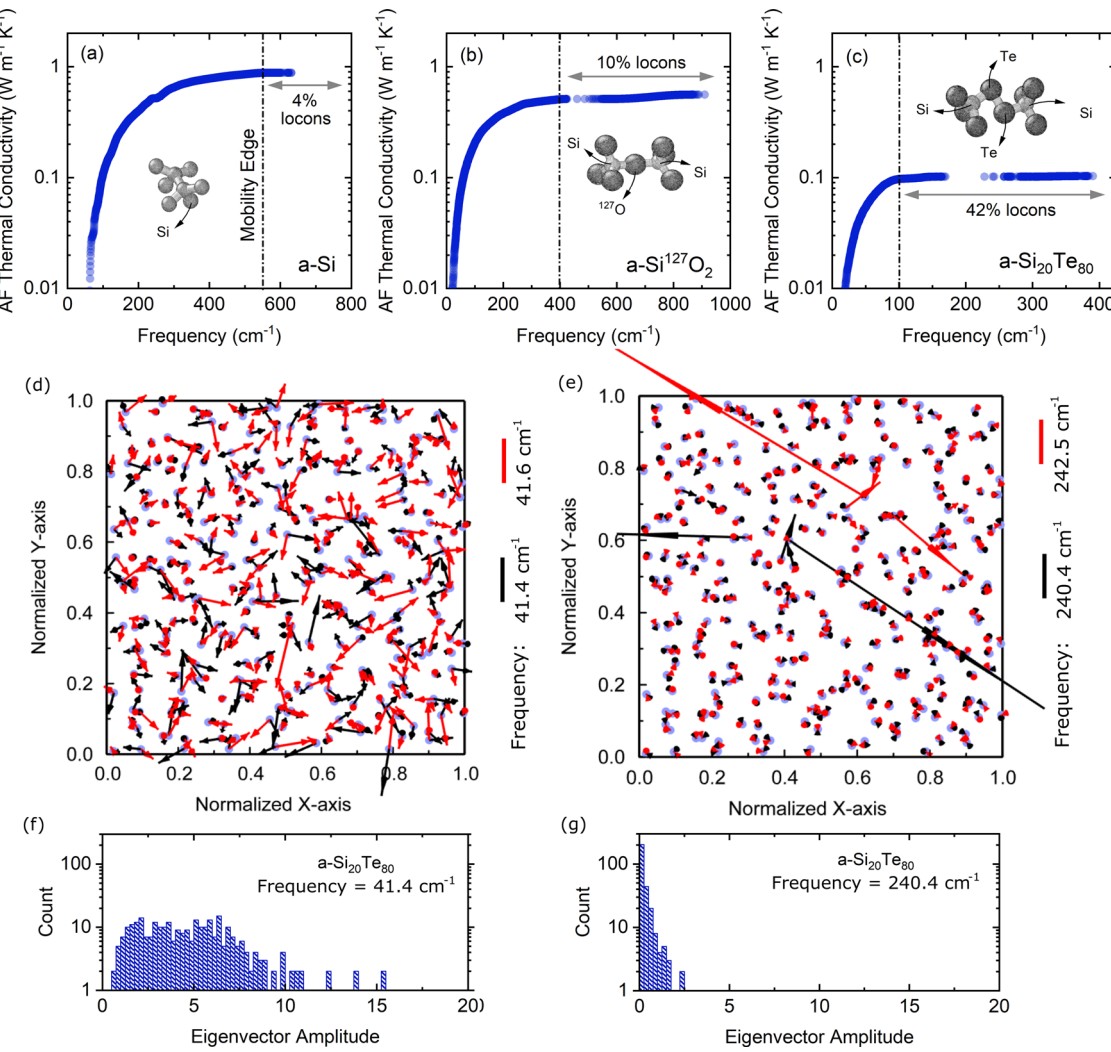

**Fig. 4 Lattice dynamics predictions for diffuson thermal conductivity and visualization of localization effects.** Allen–Feldman thermal conductivity accumulation as a function of vibrational modes frequency for **a** amorphous silicon, $a$-Si, with inset showing Si-Si bond between two tetrahedrons, **b** amorphous silicon heavy-oxide, $a$-Si$^{127}$O$_2$, with inset showing Si-$^{127}$O-Si bond between two tetrahedrons, and **c** amorphous silicon telluride, $a$-Si$_{20}$Te$_{80}$, with inset showing Si-Te-Te-Si bond between two tetrahedrons. Projection of eigenvectors associated with each mode on a $xy$-plane, indicating the degree of spatial and energetic overlap at two consecutive frequencies, **d** delocalized: 41.4 cm$^{-1}$ (black arrows) and 41.6 (red arrows) cm$^{-1}$, and **e** localized: 240.4 cm$^{-1}$ (black arrows) and 242.5 (red arrows) cm$^{-1}$ in $a$-Si$_{20}$Te$_{80}$. The few high amplitude eigenvectors at localized frequencies are an indication of strong localization showing the energy associated with these modes is confined in a small geometric region. **f, e** Histogram indicating the population of modes based on their eigenvector amplitude at delocalized and localized frequencies.

~20 and ~10 μm on the surface of the sample, respectively. The thermal model that relates the changes in the thermoreflectivity to the thermal properties of the underlying layers requires knowledge of volumetric heat capacity, thermal conductivity of the ruthenium layer, and thicknesses of the ruthenium layer and the film of interest. We use a volumetric heat capacity of 2.96 and 1.64 MJ m$^{-3}$ K$^{-1}$ and thermal conductivity of 54 and 145 W m$^{-1}$ K$^{-1}$ for ruthenium and silicon, respectively. The thickness of each layer is determined via TEM. The thermal conductivity of the Ru layer was determined from the Wiedemann–Franz law where the electrical resistivity of the Ru layer was measured using four-point probe. The composition of each sample was determined by XRF with ±3% uncertainty.

**Simulation approach.** Owing to a lack of interatomic potentials that capture the coordination number variations across the composition range in SiTe alloy, we use $a$-Si, $a$-SiO$_2$, and $a$-Si$_{20}$Te$_{80}$ compositions for comparison. The interatomic potentials used for these compositions are Stillinger–Weber for $a$-Si, Beest-Kramer-van Santen[37,46] for $a$-Si$^{127}$O$_2$, and force constants derived from ab initio calculations for $a$-Si$_{20}$Te$_{80}$.

For our ab initio density functional calculations, we have used the projector augmented wave approach as implemented in Vienna Ab initio Simulation Package (VASP)[47,48]. The exchange and correlation energies are represented in the generalized gradient approach using the Perdue-Burke-Ernzerhof exchange–correlation functional.

Given that we are investigating glassy materials, it is important to consider a large enough supercell to approximate these disordered materials. We found that 300 atoms supercells were sufficient to limit interaction with mirror images of atoms owing to the periodic boundary conditions. For Si$_{20}$Te$_{80}$, there is strong evidence that Si prefers a tetrahedral configuration bonded to four Te atoms (SiTe$_4$)[49]. Using the Packmol routine[50] as implemented in QuantumATK[51], we create 10 different initial atomic configurations by packing 60 of these SiTe$_4$ units in a supercell. The volume of the supercell is determined by the measured mass density[52] (5.08 g/cm$^3$). Structural relaxation of the system is then performed to reach the minimum energy configuration. The energies of the final structures show some distribution, which is to be expected for a glassy material that can have numerous closely spaced metastable energy states. To examine the impact of bond arrangement on the alloy properties, we also generated supercells where the initial atomic arrangement was random and there was no explicit chemical ordering. Structural relaxation was performed on 10 different configurations for these atomic networks as well. Total energy analysis confirmed the glass model based on SiTe$_4$ tetrahedra is more energetically favorable than the random physical network. For the chemically ordered bonding network, we use the relaxed configuration with the lowest energy as the basis for subsequent ab initio MD simulations. For Si$_{20}$Te$_{80}$, we ran 8 ps and 20 ps simulation runs at $T = 500$K with a fs time step. To ensure our analysis does not suffer from any thermalization effects, we discard data from the first picosecond in our trajectory analysis. From the trajectory of the MD run, we can then calculate the vibrational DOSs for Si$_{20}$Te$_{80}$. The vibrational DOSs for a

supercell with $N$ atoms can be written as:

$$S(\nu) = \frac{2}{k_B T} \sum_{i=1}^{N} \sum_{\alpha=x,y,z} m_i s_i^\alpha(\nu), \qquad (5)$$

where $\nu$ is the frequency, $m_i$ is the mass of atom $i$, and $s_i^\alpha$ is the spectral density of atom $i$ in the Cartesian direction $\alpha$. The vibrational DOSs is determined by the sum over all $N$ atoms and over all Cartesian directions. The spectral density can be written in terms of the square of the Fourier transform of the atomic velocity[53]:

$$s_i^\alpha(\nu) = \frac{\left| \int_0^{t_{max}} dt' v_i^\alpha(t') e^{-i2\pi\nu t'} \right|^2}{t_{max}}, \qquad (6)$$

where $t_{max}$ is the maximum time in the MD trajectory. The current approach is efficient and can be shown to be equivalent to the more computationally intensive calculation of the vibrational DOSs based on the velocity autocorrelation function.

Although MD simulations can provide insight into the vibrational DOSs at a given temperature, they can not provide us with direct information on the vibrational eigenvector and the degree of localization for each mode. To complement the MD simulations, we have also performed real-space force constant calculations on the fully relaxed 300 atom supercell for $Si_{20}Te_{80}$ using VASP to determine the forces on perturbed supercells (0.01 Å displacement) and Phonopy to extract the force constants and dynamical matrix[54]. For our amorphous $Si_{20}Te_{80}$ supercell, the lack of any crystalline symmetry leads to 1800 separate supercell calculations to determine the appropriate forces. The vibrational DOSs was calculated using the dynamical matrix and show good agreement with the predicted vibrational DOS from the long 20 ps MD simulation (See Supplementary Fig. 25). The accuracy of the vibrational DOS based on the MD velocity autocorrelation function improves for longer simulated time periods. This is an important consistency check between the two separate approaches to predict vibrational properties.

**Reporting summary**. Further information on research design is available in the Nature Research Reporting Summary linked to this article.

## Data availability

The data that support the findings of this study are available from the corresponding author upon reasonable request.

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

## Acknowledgements

We appreciate the support from Western Digital Technologies, Inc. The first author would like to thank Rouzbeh Rastgar, Jason Larkin, Ashutosh Giri, and Julian Gale for helpful conversations. This work was supported in part by the NSF I/UCRC on Multi-functional Integrated System Technology (MIST) Center IIP-1439644, IIP-1439680, and IIP-1738752.

## Author contributions

K.A., D.A.S., J.N., J.T.G., M.K.G., and P.E.H. designed the experiment. J.N. and J.C.R. made the samples. K.A., J.T.G., and D.H.O. performed the experiments. K.A. and D.A.S. performed the simulations. K.A., D.A.S., and P.E.H wrote the manuscript.

## Competing interests

The authors declare no competing interests.
