## [Peer Review File · Nature Communications]

REVIEWER COMMENTS

Reviewer #1 (Remarks to the Author):

Aryana et al. experimentally achieve an ultralow thermal conductivity of 0.10 ± 0.01 W/m·K in full dense amorphous SiTe. They argued that, this is a result of suppressing the thermal transport by modulating the cross-linking network among the atom. In details, they find the suppressed contribution of extended modes of vibration contribute mostly to the ultralow thermal conductivity of SiTe. This work is clearly interest to the fields of chalcogenide alloys, thus I do recommend publication in Nature Communications after major revision and addressing the following questions:

1. The author needs to highlight their novelty in the introduction. They pay much more attention to chalcogenide alloys, some other materials with low conductivity also should be introduced and analyzed, such as all-inorganic perovskite with also a reported low thermal conductivity of ~ 0.4 W/m·K. And also it is appreciated that a table should be listed to compare their thermal conductivity in the manuscript.
2. The XRD data of Te seems not so good. It is better to replace with another one. And the format of Figure 4 should be reorganized.
3. The crystal structure in Figure 1c should be re-drawn to highlight the important part.
4. The style of references should be revised in accordance with the requirement of Nature Communications.
5. The experiment section, especially the simulated information, should be explained more detailedly.

Reviewer #2 (Remarks to the Author):

This paper presented very interesting work about how vibrational modes can be localized to achieve ultralow thermal conductivities based on the experimental and simulation methods, and the result shows that the thermal conductivity of amorphous silicon telluride alloys can be suppressed to ~ 0.1 W m⁻¹ K⁻¹ via systematically tailoring the cross-linking network. The following comments/suggestions should be considered before it is accepted.

1. Why did the authors use SiTe as the research object? Please explain the innovation and practical guiding significance of this work according to the current research status.
2. The simulation model and method should be verified first.
3. Compared with other methods of measuring the thermal conductivity of thin films, what is the advantage of the TDTR method used in this manuscript? How did the authors eliminate the experimental errors (limitations of equipment)?
4. What is the electrical property of the amorphous SiTe with ultra low thermal conductivity? Whether it's going to drop dramatically as well?
5. Supplementary information, Page 5: The abbreviation "FIB" appears here for the first time, it should have the full name.
6. Supplementary information, Page 19: The authors need to check the writing on line 23.
7. Supplementary Note 2, there are several TDTR measurement signals and theoretical fits. However, the fitting curves do not start from the same delay time on the x axis. Please specify the start delay time for fitting and why that particular delay time was chosen.
8. Since the thickness of the transducer has the largest sensitivity, is it possible to adopt the value obtained from picosecond acoustics without considering its uncertainty? And the sensitivity of k_1 should be calculated too.

Reviewer #3 (Remarks to the Author):

This paper studied the thermal conductivity of amorphous chalcogenides by TDTR and MD simulation. This topic is very interesting and challenging. The minimum thermal conductivity model was found to be invalid for such amorphous materials. I appreciate the authors finding that the coordination number of atoms is of great importance for thermal transport in amorphous solids. This supports the recent model proposed by Xi et al. [Chin. Phys. Lett. 37, 104401 (2020)]. I recommend this paper to be published in Nature Communications. Some comments should be addressed:

(1) The dependence of thermal conductivity on coordination number is analyzed qualitatively by assuming that κ is proportional to $\langle r_{\text{Si}_x\text{Te}_{1-x}} \rangle \langle r_{\text{Si}} \rangle$. This assumption is intuitive. Actually, Xi et al. has pointed that $\kappa \propto r^{1/3}$. I think the authors should revisit their data with this power law dependence.

(2) The decrease of thermal conductivity in Fig. 2(a) should be partly attributed to increase of the mass density of alloy. Please mention this point in revision.

Reviewer #1 (Remarks to the Author):

Aryana et al. experimentally achieve an ultralow thermal conductivity of 0.10 ± 0.01 W/m·K in full dense amorphous SiTe. They argued that, this is a result of suppressing the thermal transport by modulating the cross-linking network among the atom. In details, they find the suppressed contribution of extended modes of vibration contribute mostly to the ultralow thermal conductivity of SiTe. This work is clearly interest to the fields of chalcogenide alloys, thus I do recommend publication in Nature Communications after major revision and addressing the following questions:

1. The author needs to highlight their novelty in the introduction. They pay much more attention to chalcogenide alloys, some other materials with low conductivity also should be introduced and analyzed, such as all-inorganic perovskite with also a reported low thermal conductivity of ~ 0.4 W/m·K. And also, it is appreciated that a table should be listed to compare their thermal conductivity in the manuscript.

Authors Response:

We appreciate the reviewer's response and suggestions. To address the reviewer's concern for highlighting the novelty of our work, we added the following statements to the introduction, page 02, paragraph 3:

“Unlike previous studies where the reduction in thermal conductivity was achieved by increasing the scattering rate of heat carriers, in this study, using different compositions of SiTe binary alloys, we demonstrate how energy transfer between vibrational modes can be dramatically suppressed, leading to strong localization and ultralow thermal conductivities.”

And later in the same paragraph:

“In contrast with previous studies, where reductions in thermal conductivity follow reductions in mass density from bond termination and networking, here, depending on the composition of the SiTe system, we achieve ultralow thermal conductivities while increasing the mass density.”

And later in the same paragraph:

“To the best of our knowledge, not only is our work the first to show how a large portion of vibrational modes can become fully localized in an amorphous medium, but it also represents the first systematic demonstration of how thermal transport changes across different topological network regimes: under-constrained, stress-free, and over-constrained.”

Also, regarding the addition of a discussion for other ultralow thermal conductivity materials, we added the following statement in the introduction, page 01, paragraph 02:

“For instance, disordered layered and cage-like crystal structures, as well as superatomic clusters with complex unit cells proved to be an effective approach in increasing the anharmonicity of the system leading to ultralow thermal conductivities. More than a decade ago, Chiritescu et al. [14] found that, by sequentially stacking extremely thin bilayers of W and Se,

the thermal conductivity of WSe₂ at room temperature dropped to a record low of ~0.05 W m⁻¹ K⁻¹, six fold lower than the corresponding minimum limit prediction.”

We need to mention that most of the ultralow thermal conductivity materials developed so far such as layered and cage structures (WSe₂ and Perovskites) possess some degree of periodicity and long-range order. In these materials, the primary heat carriers are phonons which are fundamentally different from that of the amorphous materials (propagons, diffusons, and locons). Therefore, we refrained from providing a more detailed discussion about these classes of materials in order to avoid confusion. However, per the reviewer’s request, we provided a table in the supplementary information to compare the thermal conductivity of materials with ultralow thermal conductivity:

	Composition	Structure	Thermal conductivity (W m⁻¹ K⁻¹)
Chiritescu et al. [1]	WSe ₂	layered crystalline	0.05
Duda et al. [2]	PCBM	disordered crystalline	0.055 ± 0.015
Zhang et al. [3]	Ge ₂₀ Te ₇₂ Se ₈	amorphous	0.095
This work	Si ₂₀ Te ₈₀	amorphous	0.10 ± 0.01
Rasel et al. [4]	[(C _n H _{2n+1} NH ₃) ₂ PbI ₄]	single-crystalline	0.099-1.25
Giri et al. [5]	isoBA ₂ PbI ₄	layered crystalline	0.10 ± 0.02
Zhao et al. [6]	SnSe	crystalline	0.23 ± 0.03
Lee et al. [7]	CsSnI ₃	single-crystalline	0.38 ± 0.04

References

- [1] Chiritescu, C., Cahill, D.G., Nguyen, N., Johnson, D., Bodapati, A., Keblinski, P. and Zschack, P., 2007. Ultralow thermal conductivity in disordered, layered WSe₂ crystals. *Science*, 315(5810), pp.351-353.
- [2] Duda, J.C., Hopkins, P.E., Shen, Y. and Gupta, M.C., 2013. Exceptionally low thermal conductivities of films of the fullerene derivative PCBM. *Physical review letters*, 110(1), p.015902.
- [3] Zhang, S.N., He, J., Zhu, T.J., Zhao, X.B. and Tritt, T.M., 2009. Thermal conductivity and specific heat of bulk amorphous chalcogenides Ge₂₀Te_{80-x}Se_x (x= 0, 1, 2, 8). *Journal of Non-Crystalline Solids*, 355(2), pp.79-83.
- [4] Rasel, M.A.J., Giri, A., Olson, D.H., Ni, C., Hopkins, P.E. and Feser, J.P., 2020. Chain-Length Dependence of Thermal Conductivity in 2D Alkylammonium Lead Iodide Single Crystals. *ACS Applied Materials & Interfaces*, 12(48), pp.53705-53711.
- [5] Giri, A., Chen, A.Z., Mattoni, A., Aryana, K., Zhang, D., Hu, X., Lee, S.H., Choi, J.J. and Hopkins, P.E., 2020. Ultralow thermal conductivity of two-dimensional metal halide perovskites. *Nano letters*, 20(5), pp.3331-3337.
- [6] Zhao, L.D., Lo, S.H., Zhang, Y., Sun, H., Tan, G., Uher, C., Wolverton, C., Dravid, V.P. and Kanatzidis, M.G., 2014. Ultralow thermal conductivity and high thermoelectric figure of merit in SnSe crystals. *Nature*, 508(7496), pp.373-377.
- [7] Lee, W., Li, H., Wong, A.B., Zhang, D., Lai, M., Yu, Y., Kong, Q., Lin, E., Urban, J.J., Grossman, J.C. and Yang, P., 2017. Ultralow thermal conductivity in all-inorganic halide perovskites. *Proceedings of the National Academy of Sciences*, 114(33), pp.8693-8697.

2. The XRD data of Te seems not so good. It is better to replace with another one. And the format of Figure 4 should be reorganized.

Authors Response:

We appreciate the reviewer's request for additional higher quality XRD results. We have performed XRD measurements using three different systems at three different locations on various samples (with and without Ru transducer) in order to ensure the accuracy of the measurements. These results have been provided in the supplementary information and are in good agreement with prior literature observations. In addition, we need to note that the Te film is only 30 nm thick which makes it quite challenging in terms of characterization. The XRD measurement results are given as follows:

Figure 1. Grazing incidence XRD measurements across different compositions of SiTe coated with 80 nm of Ruthenium on a silicon substrate.

Figure 2. Comparison between X-ray diffraction for high Te content samples. The results presented here are for samples with Si substrate and 80 nm specular Ru coating.

Figure 3. Comparison between X-ray diffraction in this study and literature. (a) XRD results are from samples with SiO_2 substrate and 3-5 nm carbon coating, (b) XRD results are from samples with Si substrate and 80 nm specular Ru coating.

3. The crystal structure in Figure 1c should be re-drawn to highlight the important part.

Authors Response:

We appreciate the reviewer's comment regarding the edition of Figure 1c. We changed the figure and its caption to the following in order to emphasize on the important structure:

Figure 4. (a) Schematics of amorphous networks with a total of 15 atomic sites for different Te to Si ratios representing various topological regimes: (i) under-constrained network, (ii) stress-free network, and (iii) over-constrained network. (b) Raman spectra for SiTe at different Te concentrations. The sharp silicon peak is the effect of the substrate and is not associated with the thin film properties. (c) Illustration of the TDTR measurement, a representative atomic structure of α -Si₂₀Te₈₀ with 300 atoms forming randomly oriented tetrahedrons where each Si is bonded to four Te atoms, and a TEM showing the relevant layers in α -Si₁₉Te₈₁. (d) Representative TDTR data for 40 nm thick α -Si₁₉Te₈₁. The inset shows the picosecond acoustic measurements where the distance between the troughs and the peaks in the measured TDTR data are related to the time it takes for acoustic waves traveling at the speed of sound to traverse the α -Si₁₉Te₈₁ film.

4. The style of references should be revised in accordance with the requirement of Nature Communications.

Authors Response:

We appreciate the reviewer's comment. We reformatted the manuscript and the references to comply with the Nat. Comms. guidelines.

5. The experiment section, especially the simulated information, should be explained more detailedly.

Authors Response:

We appreciate the reviewer's request for expanding on the experimental and simulation section. In this regard, we would like to mention that in the Supplementary Note 2, we have provided extensive discussion regarding the experimental approach, the thermal model, and the sensitivity of our measurements to key parameters such as layers thickness, heat capacity, and thermal conductivity. In order to address the reviewer's concern, we added the following details to the main manuscript, experimental section, page 08, last paragraph:

“The thermal conductivity of the Ru layer was determined from the Wiedemann-Franz law where the electrical resistivity of the Ru layer was measured using 4-point probe. The composition of each sample was determined by x-ray fluorescence (XRF) with $\pm 3\%$ uncertainty.”

Furthermore, many of the details of the simulated information were included in the supplementary information due to space constraints. In Supplementary Note 3, we provide a detailed discussion of the complementary equilibrium and non-equilibrium molecular dynamics approach used to investigate the effect of mass scattering on the thermal conductivity of a-SiTe. The use of the two techniques provided an important internal consistency check. In Supplementary Note 5, we provide a full description of the Allen-Feldman formalism for predicting diffuson thermal conductivity and a technical discussion of the associated force constant calculations to determine key parameters (e.g, eigenvectors, mode heat capacity, and mode diffusivity).

However, we agree that it would be helpful to provide more details in the main text on the Allen-Feldman approach used to calculate diffuson thermal conductivity. To this end, we have added the following sentence in the main text, page 04, last paragraph:

“According to AF theory, the thermal conductivity due to diffuson contribution can be approximated by:

$$\kappa_{AF} = \frac{1}{V} \sum_{i=1}^N C(\omega_i) D(\omega_i) \quad (2)$$

where V is the system volume, ω_i is frequency of i th mode, and $C(\omega_i)$ and $D(\omega_i)$ are the frequency dependent specific heat and modes diffusivity, respectively.”

Reviewer #2 (Remarks to the Author):

This paper presented very interesting work about how vibrational modes can be localized to achieve ultralow thermal conductivities based on the experimental and simulation methods, and the result shows that the thermal conductivity of amorphous silicon telluride alloys can be suppressed to $\sim 0.1 \text{ W m}^{-1} \text{ K}^{-1}$ via systematically tailoring the cross-linking network. The following comments/suggestions should be considered before it is accepted.

1. Why did the authors use SiTe as the research object? Please explain the innovation and practical guiding significance of this work according to the current research status.

Authors Response:

We would like to thank the reviewer for raising this fundamental question. In recent years, we have entered a new era where glasses and non-crystalline solids have found unprecedented applications from medical to semiconductor industries due to their remarkable properties [1]. Despite the ubiquity of glasses in our everyday life, the fundamental physics behind this class of materials have not been properly realized [2]. This has rekindled interests in the glass community to better understand the properties of amorphous structures to discover more exotic and efficient materials. Amorphous chalcogenides, in particular, have been under extensive research in the fields of thermoelectrics and data storage devices [2-5]. Although there have been a number of different studies regarding the mechanical [6,7], optical [8-9], and electrical [10,11] properties of amorphous materials, the thermal properties of these materials are yet to be unveiled. In this paper, for the first time, we demonstrate how thermal transport changes across different topological constraint regimes, i.e., under-constrained, stress-free, and over-constrained. In addition, for the first time, we demonstrated a significant reduction in thermal conductivity with a mechanism (localization) other than scattering or shortening the mean free paths of the energy carriers. The following sentences have been added to the introduction on page 02 paragraph 3:

“Unlike previous studies where the reduction of thermal conductivity is achieved through increasing the scattering rate of heat carriers, in this study, using different compositions of SiTe binary alloys, we demonstrate how vibrational modes can become fully localized to achieve ultralow thermal conductivities.”

And later in the same paragraph:

“To the best of our knowledge, not only is our work the first to show how vibrational modes can become fully localized in an amorphous medium, but it also represents the first systematic demonstration of how thermal transport changes across different topological network regimes: under-constrained, stress-free, and over-constrained.”

In order to investigate properties of glasses, traditionally, covalently bonded elements such as Si, Ge, As, and Sb, as well as chalcogens such as S, Se and Te have been the subject of research. The difference between the coordination number of elements like Si and Te provides an excellent testbed to study the effect of topological constraint in amorphous solids. Although elements in group V such as arsenic and antimony have coordination number higher than chalcogen elements

and could be another suitable candidate for this study, we refrained from choosing them not only because they are toxic but also, because they provide a narrower range of coordination number $2 < \langle r \rangle < 3$ with respect to their composition compared to group IV elements $2 < \langle r \rangle < 4$. Another important reason behind choosing Si and Te from other existing possibilities such as Ge and Se is because they provide the largest mass difference (~ 5 times) which leads to higher scattering rate of energy carriers and potentially further reduction of thermal conductivity. We studied the effect of mass scattering in SeTe and observed that the thermal conductivity barely changes in SeTe at different Te concentrations. This result is expected considering the fact that Se and Te are both heavy elements with masses relatively close to each other and constant coordination number.

As a consequence, SiTe seemed the most promising composition not only because it enables us to capture thermal properties in different topological regime, but also the large atomic mass mismatch between Si and Te will be an additional source of scattering which will further reduce the thermal conductivity.

References

- [1] Morse, D.L. and Evenson, J.W., 2016. Welcome to the glass age. *International Journal of Applied Glass Science*, 7(4), pp.409-412.
- [2] Wingert, M.C., Zheng, J., Kwon, S. and Chen, R., 2016. Thermal transport in amorphous materials: a review. *Semiconductor Science and Technology*, 31(11), p.113003.
- [3] He, M., Zhao, Y., Wang, B., Xi, Q., Zhou, J. and Liang, Z., 2015. 3D printing fabrication of amorphous thermoelectric materials with ultralow thermal conductivity. *Small*, 11(44), pp.5889-5894.
- [4] Hirose, Y., Tsuchii, M., Shigematsu, K., Kakefuda, Y., Mori, T. and Hasegawa, T., 2019. Thermoelectric properties of amorphous ZnO_xN_y thin films at room temperature. *Applied Physics Letters*, 114(19), p.193903.
- [5] Hosseini, P., Wright, C.D. and Bhaskaran, H., 2014. An optoelectronic framework enabled by low-dimensional phase-change films. *Nature*, 511(7508), pp.206-211.
- [6] Wright, C.D., Liu, Y., Kohary, K.I., Aziz, M.M. and Hicken, R.J., 2011. Arithmetic and biologically-inspired computing using phase-change materials. *Advanced Materials*, 23(30), pp.3408-3413.
- [7] He, H. and Thorpe, M.F., 1985. Elastic properties of glasses. *Physical Review Letters*, 54(19), p.2107.
- [8] Nordell, B.J., Nguyen, T.D., Caruso, A.N., Lanford, W.A., Henry, P., Li, H., Ross, L.L., King, S.W. and Paquette, M.M., 2019. Topological Constraint Theory Analysis of Rigidity Transition in Highly Coordinate Amorphous Hydrogenated Boron Carbide. *Frontiers in Materials*, 6, p.264.
- [9] Tichý, L., Ticha, H., Nagels, P., Callaerts, R., Mertens, R. and Vlček, M., 1999. Optical properties of amorphous As–Se and Ge–As–Se thin films. *Materials Letters*, 39(2), pp.122-128.
- [10] Halimah, M.K., Daud, W.M., Sidek, H.A.A., Zaidan, A.W. and Zainal, A.S., 2010. Optical properties of ternary tellurite glasses. *Mater. Sci. Pol*, 28(1), pp.173-180.
- [11] Nordell, B.J., Karki, S., Nguyen, T.D., Rulis, P., Caruso, A.N., Purohit, S.S., Li, H., King, S.W., Dutta, D., Gidley, D. and Lanford, W.A., 2015. The influence of hydrogen on the chemical, mechanical, optical/electronic, and electrical transport properties of amorphous hydrogenated boron carbide. *Journal of Applied Physics*, 118(3), p.035703.

[12] Nomura, K., Kamiya, T. and Hosono, H., 2012. Effects of diffusion of hydrogen and oxygen on electrical properties of amorphous oxide semiconductor, In-Ga-Zn-O. ECS Journal of Solid State Science and Technology, 2(1), p.P5.

2. The simulation model and method should be verified first.

Authors Response:

We agree that it is always essential to carefully evaluate the simulation model and methods used in any research effort.

In this work, we first explored a simple analytical model for the minimum diffuson thermal conductivity in a material with varying mean coordination. This model builds on the previous minimum thermal conductivity model proposed by Agne *et al.* [1] and takes into account the impact of atom connectivity. Overall, it is able to reproduce our measured trend with thermal conductivity for the SiTe alloys quite well. This simple model captures the key diffuson physics and should prove useful for exploring trends in the thermal conductivity of other chalcogenide alloys.

In order to verify this analytical diffuson model and delve deeper into the microscopic heat transfer, we used a variety of robust and predictive simulation approaches (molecular dynamics, density functional theory, lattice dynamics) that have been successfully used to model similar materials in the past. In the paper, we have highlighted these techniques in the discussion, and we have also provided relevant citations for the interested reader. Given the importance of the Allen-Feldman approach to predicting diffuson thermal conductivity, we have added the AF equation used for this calculation in the main text, page 04, last paragraph:

“According to AF theory, the thermal conductivity due to diffuson contribution can be approximated by:

$$\kappa_{AF} = \frac{1}{V} \sum_{i=1}^N C(\omega_i) D(\omega_i) \quad (2)$$

where V is the system volume, ω_i is frequency of i th mode, $C(\omega_i)$ and $D(\omega_i)$ are the frequency dependent specific heat and mode diffusivity, respectively.”

and, for the sake brevity, the details of this approach are discussed in the Supplementary Note 5. We also added the following to the Supplementary Note 5, page 30, 1st paragraph:

“To obtain the AF thermal conductivity, we developed our own script using MATLAB to perform the above calculations. For this, we imported the dynamical matrix, eigenvector, position, and frequency of the modes from the force constant calculations into our model and estimated the diffusivity and thermal conductivity. It is worthwhile mentioning that there is a program, General Utility Lattice Program (GULP), which automatically performs the AF thermal conductivity calculations for a limited number of potentials such as harmonic, Lennard-Jones, and Stillinger-Weber (SW) to name a few. However, since we did not use any of the existing potentials for calculating the thermal conductivity of α -Si₂₀Te₈₀, we had to write our own script. Nonetheless, we used GULP to validate our script by calculating the diffusivity and AF thermal conductivity of α -Si and α -SiO₂ and comparing the results with those obtained by GULP.”

And added the following to the page 30, 2nd paragraph:

“Allen and Feldman recommended a broadening factor larger than the average frequency spacing $\delta\omega_{ave}$. Considering this, we observe changing the broadening factor from $1\delta\omega_{ave}$ =

0.4360 cm⁻¹ to $5\delta\omega_{ave} = 2.1801 \text{ cm}^{-1}$ leads to small increase in the estimated thermal conductivity (~3%).”

In addition, we have also performed several internal checks to confirm that using different simulation techniques provide us with the same predicted values. These technical discussions and internal checks have been included in the supplementary materials, but we briefly highlight some of this analysis below.

In Supplementary Note 3 where we examine the impact of mass scattering on the thermal conductivity of a-SiTe, we use both equilibrium Green-Kubo and non-equilibrium molecular dynamics approaches to predict thermal conductivity. As noted in the supplementary material, both approaches result in the same predicted thermal conductivity.

For the prediction of the vibrational density of states in Si₂₀Te₈₀, we calculate this using the velocity autocorrelation function from a long molecular dynamics simulation and also from a direct supercell force constant calculation. Both approaches lead to vibrational density of states in good agreement providing an additional check on the vibrational properties used to predict the diffuson thermal conductivity. The use of the two techniques provided an important internal consistency check.

For the prediction of diffuson thermal conductivity using the Allen-Feldman formalism, as noted in Supplementary Note 5, our predicted value for amorphous silicon is comparable with the previous study by Larkin and McGaughey [2].

References

- [1] Agne, M.T., Hanus, R. and Snyder, G.J., 2018. Minimum thermal conductivity in the context of diffuson-mediated thermal transport. *Energy & Environmental Science*, 11(3), pp.609-616.
- [2] Larkin, J.M. and McGaughey, A.J., 2014. Thermal conductivity accumulation in amorphous silica and amorphous silicon. *Physical Review B*, 89(14), p.144303.

3. Compared with other methods of measuring the thermal conductivity of thin films, what is the advantage of the TDTR method used in this manuscript? How did the authors eliminate the experimental errors (limitations of equipment)?

Authors Response:

This is a fair question. In fact, there have been other thermal conductivity reports near the rigidity threshold with different thermometry techniques such as differential scanning calorimetry and transient plane source method [1,2]. In these studies, they observe a maximum at the rigidity threshold which is in contrast to what we observe (minimum at the rigidity threshold and under-constrained regime). Considering the fact that Ref [1] reports a thermal diffusivity for bulk a-Si₂₀Te₈₀ which is a factor 2 higher than amorphous silicon, it is possible that the samples they studied have not been entirely amorphous or a large concentration of impurities existed in their sample. The discrepancy between the thin film results and bulk material, however, is not surprising as the defects such as impurities and inhomogeneity are common in bulk systems.

One of the major challenges in making amorphous materials is to make the material uniformly amorphous across the entire volume while maintaining the pristine composition. This is even a bigger challenge when the size of the material under study is large (bulk). However, when depositing amorphous materials as thin films, it is easier to ensure the uniformity and purity of the system. In this respect, one of the main advantages of TDTR compared to other existing thermometry technique such as laser flash is that it can measure thermal properties of thin films and buried surfaces. In addition, the transducer coating prevents the film from oxidation and maintains the film in the pristine state.

In order to ensure the accuracy of our results, we performed TDTR measurements on two different thickness series that have been deposited in different runs with different adjacent materials:

- 1- Ru / **5 nm CN_x** / 5-40 nm Si₂₀Te₈₀ / **5 nm CN_x** / Si substrate
- 2- Ru / 5-40 nm Si₂₀Te₈₀ / **5 nm W** / Si substrate

These samples then have been measured on two different TDTR set up with different pump beam wavelengths (808 nm 408 nm). These results are presented in Supplementary Fig. 17 (d,e) which shows excellent agreement between different sample geometries.

References

- [1] Philip, J. and Madhusoodanan, K.N., 1988. Percolation threshold of thermal conduction in A_xIV B_{1-x}VI chalcogenide semiconducting glasses. *Physical Review B*, 38(6), p.4127.
- [2] Lonergan, J., Smith, C., McClane, D. and Richardson, K., 2016. Thermophysical properties and conduction mechanisms in A_{Sx}Se_{1-x} chalcogenide glasses ranging from x= 0.2 to 0.5. *Journal of Applied Physics*, 120(14), p.145101.

4. What is the electrical property of the amorphous SiTe with ultralow thermal conductivity? Whether it's going to drop dramatically as well?

Authors Response:

We would like to thank the reviewer for asking if electrons contribute to thermal conductivity of the SiTe. The short answer is that there is no contribution from the electron in the unperturbed state. For addressing the reviewer's concern, we added the following as a supplementary note 6, in page 35 of the supplementary materials:

"A common approach to estimate the thermal conductivity due to electron contribution is the widely used empirical equation proposed by Wiedemann-Franz (WF):

$$k = k_p + k_e$$
$$k_e = \frac{LT}{\rho}$$

where k_p and k_e are thermal conductivities due to phonon and electron contribution, respectively, L is the Lorenz number $2.44 \times 10^{-8} \text{ W } \Omega \text{ K}^{-2}$, T is temperature, and ρ is the electrical resistivity. According to Bailey [1] and Petersen et al. [2], the electrical resistivity of the $\text{Si}_x\text{Te}_{1-x}$ for $0.02 < x < 0.4$ is in the range of $0.1\text{-}5000 \text{ } \Omega\text{m}$. We confirmed that for the $\text{Si}_{20}\text{Te}_{80}$ composition the electrical resistivity in our samples is at least 100 Ohm m , consistent with this range. This corresponds to electronic contribution, k_e , of $7 \times 10^{-5} - 1 \times 10^{-9} \text{ W/m/K}$ which is orders of magnitude lower than the measured thermal conductivity. This indicates that SiTe is electrically insulating in its room temperature unperturbed state. This is also confirmed by a more recent study [3] which showed threshold switching behavior of SiTe with resistance before switching as high as ($> 1 \text{ G}\Omega$ at 0.1 V). The high resistivity of $a\text{-SiTe}$ is attributed to the existence of deep trap states."

References

- [1] Bailey, L.G., 1966. Preparation and properties of silicon telluride. Journal of Physics and Chemistry of Solids, 27(10), pp.1593-1598.
- [2] Petersen, K.E., Birkholz, U. and Adler, D., 1973. Properties of crystalline and amorphous silicon telluride. Physical Review B, 8(4), p.1453.
- [3] Koo, Y., Lee, S., Park, S., Yang, M. and Hwang, H., 2017. Simple binary ovonic threshold switching material SiTe and its excellent selector performance for high-density memory array application. IEEE Electron Device Letters, 38(5), pp.568-571.

5. Supplementary information, Page 5: The abbreviation "FIB" appears here for the first time, it should have the full name.

Authors Response:

We would like to thank the reviewer for pointing out to this. We added the full name for FIB in its first appearance in the Supplementary Note 1, page 05:

"For this purpose, cross section view Focused Ion Beam (FIB) chips were prepared from the center of each wafer piece."

6. Supplementary information, Page 19: The authors need to check the writing on line 23.

Authors Response:

We would like to thank the reviewer for pointing out to the error in the figure number. We fixed this issue in the updated version of the manuscript.

7. Supplementary Note 2, there are several TDTR measurement signals and theoretical fits. However, the fitting curves do not start from the same delay time on the x axis. Please specify the start delay time for fitting and why that particular delay time was chosen.

Authors Response:

We would like to thank the reviewer for pointing this. For analyzing the TDTR data, we normally start the fit from delay time of 300 ps. However, in Supplementary Fig. 17 (c), the TDTR fit starts from 1000 ps. After revisiting the data and fitting from delay time of 300 ps we observe less than 1% change in fitted parameter (thermal conductivity). This is consistent with the sensitivity analysis presented in the same figure (also given in the next page) where the sensitivity to the parameters of interest does not change from 300 ps to 1000 ps. We updated the figure and started the fit from 300 ps for consistency.

8. Since the thickness of the transducer has the largest sensitivity, is it possible to adopt the value obtained from picosecond acoustics without considering its uncertainty? And the sensitivity of k_{-1} should be calculated too.

Authors Response:

We would like to thank the reviewer for raising this question. This concern arises from a statement we originally had in the Supplementary Note 2:

“we are able to measure the thickness of the Ru transducer via picosecond acoustics”

However, this statement was made in the earlier stages of drafting the paper when we did not have extensive TEM characterizations. In the light of TEM micrographs which we have for nearly all of the samples studied here, we did not have to measure the Ru transducer from picosecond acoustics. In addition, after observing some non-uniformity in the thickness of some of the films such as Te in TEM micrographs, we calculate the uncertainty based on the thickness of the film. The above statement is removed from the updated version of SI to avoid confusion.

Also, regarding the sensitivity of our measurement to transducer thermal conductivity, please note that, due to the ultralow thermal conductivity of the film, we are mostly sensitive to in-plane thermal conductivity of the transducer rather than the cross-plane. Therefore, the sensitivity for the in-plane thermal conductivity of the transducer, $k_{1(\text{in-plane})}$, is calculated and depicted in the following figure, which still shows negligible effect in our measurements for both films thickness.

Figure 5. Sensitivity of our measurement to thermal conductivity, k , and thermal boundary conductance, G , for (a) 5 and (b) 40 nm thick $\text{Si}_{20}\text{Te}_{80}$, (c) a representative fit to the experimental data for a 5 and 40 nm thick $\text{Si}_{20}\text{Te}_{80}$. (d,e) thermal conductivity of $\text{Si}_{20}\text{Te}_{80}$ for different samples configurations, obtained by applying a linear fit to the thermal resistance due to each thickness. The solid triangles correspond to uncertainty and is calculated by assuming 10% variations in the $\text{Si}_{20}\text{Te}_{80}$ film thickness.

Reviewer #3 (Remarks to the Author):

This paper studied the thermal conductivity of amorphous chalcogenides by TDTR and MD simulation. This topic is very interesting and challenging. The minimum thermal conductivity model was found to be invalid for such amorphous materials. I appreciate the authors finding that the coordination number of atoms is of great importance for thermal transport in amorphous solids. This supports the recent model proposed by Xi et al. [Chin. Phys. Lett. 37, 104401 (2020)]. I recommend this paper to be published in Nature Communications. Some comments should be addressed:

(1) The dependence of thermal conductivity on coordination number is analyzed qualitatively by assuming that P is proportional to $\langle r_{\text{Si}_x\text{Te}_{1-x}} \rangle / \langle r_{\text{Si}} \rangle$. This assumption is intuitive. Actually, Xi et al. has pointed that $\kappa \propto r^{1/3}$. I think the authors should revisit their data with this power law dependence.

Authors Response:

We would like to thank the reviewer for recommending this related paper to estimate the thermal conductivity. We revised our model by using the suggested power law dependence in this paper, $k \propto P^{1/3}$, where $P = \langle r_{\text{alloy}} \rangle / \langle r_{\text{Si}} \rangle$. Using this assumption, our model shows a better agreement with experimental results. The following sentences have been added to the manuscript page 5, paragraph 2:

“In this case, the calculation of $\kappa_{\text{min,D}}$ assumes that P changes with respect to the mean coordination number of each composition. Following the model proposed by Xi et al. [29] showing a power law dependence between mean coordination number and thermal conductivity ($k \propto \langle r_m \rangle^{1/3}$), we normalize each coordination number with respect to pure Si, i.e., $P = (\langle r_m \rangle / \langle r_{\text{max}} \rangle)^{1/3} = (\langle r_{\text{SiTe}} \rangle / \langle r_{\text{Si}} \rangle)^{1/3}$. Using this assumption, P changes from 1 for Si ($\langle r_{\text{Si}} \rangle = 4$) to ~ 0.8 for Te ($\langle r_{\text{Te}} \rangle = 2$). Applying this condition, we calculate the diffuson-mediated thermal conductivity for $\text{Si}_{19}\text{Te}_{81}$ as $0.12 \text{ W m}^{-1} \text{ K}^{-1}$, in better agreement with the measured values for our SiTe alloys (see supplementary note 4).”

(2) The decrease of thermal conductivity in Fig. 2(a) should be partly attributed to increase of the mass density of alloy. Please mention this point in revision.

Authors Response:

We would like to thank the reviewer for pointing out to this. We added following statement to page 4, paragraph 2:

“Although the increased mass density of high content Te alloys and the large atomic mass mismatch between Si and Te could partly explain the reductions observed in thermal conductivity, we show they are not the primary reason behind such a dramatic reduction in thermal conductivity.”

REVIEWERS' COMMENTS

Reviewer #2 (Remarks to the Author):

The scientific and technical content of the revised manuscript is consistency. The importance and originality of the revised manuscript could meet the requirements of this journal. The revised manuscript is suggested to be published in the journal.

Reviewer #3 (Remarks to the Author):

The authors have addressed all my concerns in the revised manuscript. It can be accepted for publication.